# Abiotic synthesis of graphitic carbons in the Eoarchean Saglek-Hebron metasedimentary rocks

Zixiao Guo [1,2] ✉, Dominic Papineau [2,3,4,5], Jonathan O'Neil[6], Hanika Rizo[7], Zhong-Qiang Chen [5], Xincheng Qiu[5] & Zhenbing She [5]

Graphite in metasedimentary rocks of the Eoarchean Saglek-Hebron Gneiss Complex (Canada) is depleted in $^{13}C$ and has been interpreted as one of the oldest traces of life on Earth. The variation in crystallinity of this oldest graphitic carbon could possibly confirm the effect of metamorphism on original biomass, but this is still unexplored. Here, we report specific mineral associations with graphitic carbons that also have a range of crystallinity in the Saglek-Hebron metasedimentary rocks. Petrographic, geochemical and spectroscopic analyses in the Saglek-Hebron banded iron formations suggest that poorly crystalline graphite is likely deposited from C-H-O fluids derived from thermal decomposition of syngenetic organic matter, which is preserved as crystalline graphite during prograde metamorphism. In comparison, in the Saglek-Hebron marble, disseminations of graphite co-occur with carbonate and magnetite disseminations, pointing to abiotic synthesis of graphitic carbons via decarbonation. Our results thus highlight that variably crystalline graphitic carbons in the Saglek-Hebron metasedimentary rocks are potential abiotic products on early Earth, which lay the groundwork for identifying the preservation of prebiotic organic matter through metamorphism on Earth and beyond.

The search for evidence of early life is a challenging problem owing to the highly metamorphosed nature of the oldest sedimentary rocks containing organic matter[1,2]. Metamorphism is known to distort the morphology of microbial remains through the alteration of organic matter to crystalline graphite, and purported microfossils from highly metamorphosed rocks have been reported from the Nuvvuagittuq Supracrustal Belt in northern Canada (3.75–4.3 Ga (gigayears ago))[3,4], which has been questioned in origin[5]. This requires a broader search for direct evidence of life during the Eoarchean (3.6–4.0 Ga) to also include chemical and isotopic biosignatures, which in this context

should be defined as the possible signatures of life. Biosignatures of graphitic carbons in >3.6 Ga rocks have thus been reported in the southwest Greenland Isua Supracrustal Belt (>3.7 Ga)[6–8] and the Akilia association (>3.83 Ga)[9–11], and in the northeast Canada Saglek-Hebron Gneiss Complex (~3.9 Ga)[12]. Observations show high-grade metamorphism results in biomass losing heteroatoms such as H, N, O and S[13] as well as $^{12}C$ and $^{14}N$ isotopes, thereby leaving isotopically heavy residual graphitic carbons[14]. To determine the precursor carbon in such graphitic carbon requires consideration of the effects of metamorphism and various abiotic processes that could potentially explain

[1]School of Geographical Sciences, Hebei Key Laboratory of Environmental Change and Ecological Construction, Hebei Normal University, Shijiazhuang, China. [2]Department of Earth Sciences, University College London, London, UK. [3]London Centre for Nanotechnology, University College London, London, UK. [4]Centre for Planetary Sciences, University College London, London, UK. [5]State Key Laboratory of Biogeology and Environmental Geology, China University of Geosciences (Wuhan), Wuhan, PR China. [6]Department of Earth and Environmental Sciences, Ottawa-Carleton Geoscience Centre, University of Ottawa, Ottawa, ON, Canada. [7]Department of Earth Sciences, Ottawa-Carleton Geoscience Centre, Carleton University, Ottawa, ON, Canada. ✉e-mail: zxguo@hebtu.edu.cn

their composition and crystallinity in metamorphosed sedimentary rocks[1,15].

The Saglek-Hebron Gneiss Complex is a granite-greenstone terrane located in the west end of the North Atlantic Craton along the east coast of northern Labrador, Canada (Supplementary Fig. 1). It is mostly composed of orthogneisses from the trondhjemite-tonalite-granodiorite (TTG) series, as old as ~3.9 Ga TTG[16,17], but also includes basaltic metavolcanic rocks interbedded with clastic and chemical metasediments[18]. It is acknowledged that the Saglek-Hebron Gneiss Complex includes multiple generation of TTG gneisses with ages from Eoarchean to Neoarchean[17,19], with perhaps two distinct assemblages of Eoarchean to Paleoarchean supracrustal rocks[20,21], which have succumbed to a complex and protracted thermal history (see the Supplementary Information; Supplementary Fig. 1)[22]. Among them, the oldest volcano-sedimentary assemblage, referred to as Nulliak, is interpreted to have an age of nearly 3.8 Ga[18,21,23]. Tashiro et al. (2017) reported [13]C-deleted graphite in some metasedimentary rocks from the Saglek-Hebron Gneiss Complex and interpreted these occurrences as one of the oldest traces of life on Earth[12]. An increasing number of studies raised doubts about the interpretation of the age and origin of graphitic carbons in these metasedimentary rocks, suggesting that such graphitic carbons could have precipitated from hydrothermal and metasomatic fluids during post-Eoarchean metamorphic episodes[24,25]. Hence, in this context, the distinction between abiotic and biological processes and sources of organic molecules in graphitic carbons is an imperative goal to achieve accurate identification of traces of early life on Earth.

To address this outstanding problem, correlative petrographic, geochemical and spectroscopic analyses are used to probe specific mineral associations with graphitic carbons in the Eoarchean Saglek-Hebron metasedimentary rocks. Our results suggest that graphite in the Saglek-Hebron banded iron formation (BIF) is likely deposited from C-H-O fluids, while graphite associated with magnetite and carbonate in the Saglek-Hebron marble arises from decarbonation. Hence, the new observations provide a documentation of abiotic transformation of primitive sedimentary organic matter on early Earth, which has implications for identifying early microbial remnants, as well as for understanding geochemical cycling of carbon on early Earth.

## Results and discussion
### Characteristics of graphite in the Saglek-Hebron metasedimentary rocks

Graphitic carbons were studied in three samples from the Eoarchean Nulliak supracrustal assemblage of the Saglek-Hebron Gneiss Complex located in northern Labrador, Canada (Fig. 1a–c). Sample *SG-274* is a BIF dominated by grunerite, quartz and variable-sized magnetite, with trace amounts of organic matter, hastingsite, actinolite, apatite, calcite, chlorite, and biotite, whereas the BIF sample *SG-236* is dominated by pyroxene + quartz + magnetite (Fig. 1d, e and Supplementary Data 1). In comparison, sample *SG-275* is dominated by carbonate +

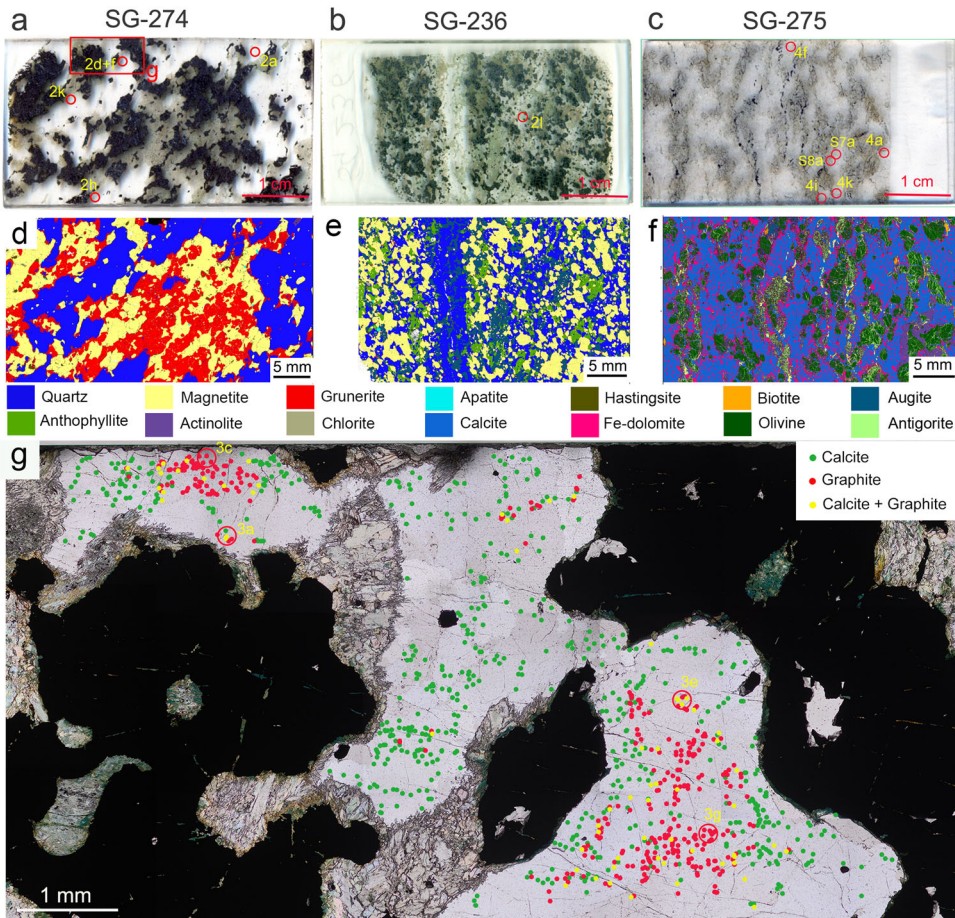

**Fig. 1 | Thin section scans of the Saglek-Hebron metasedimentary rocks in this study.** The thin sections **a** SG-274 (banded iron formation; BIF), **b** SG-236 (BIF), and **c** SG-275 (marble) are shown with a width of 2.5 cm. Corresponding X-ray mineral maps of thin sections **d** SG-274, and **e** SG-236 showing a dominance of quartz, magnetite, and amphiboles, and **f** SG-275 dominated by calcite, ankerite, olivine, and antigorite. **g** Selected area in SG-274 (red box in panel **a**) showing 576 calcite grains (green dots), 339 graphite grains (Gra$_I$; red dots), and 64 calcite + graphite associations (Gra$_{II}$; yellow dots). The petrographic mode of occurrence shows significantly more Gra$_I$ and Gra$_{II}$ near the central zone of coarse quartz and mostly graphite-free calcite grains near the Fe-silicate and magnetite portions of the BIF.

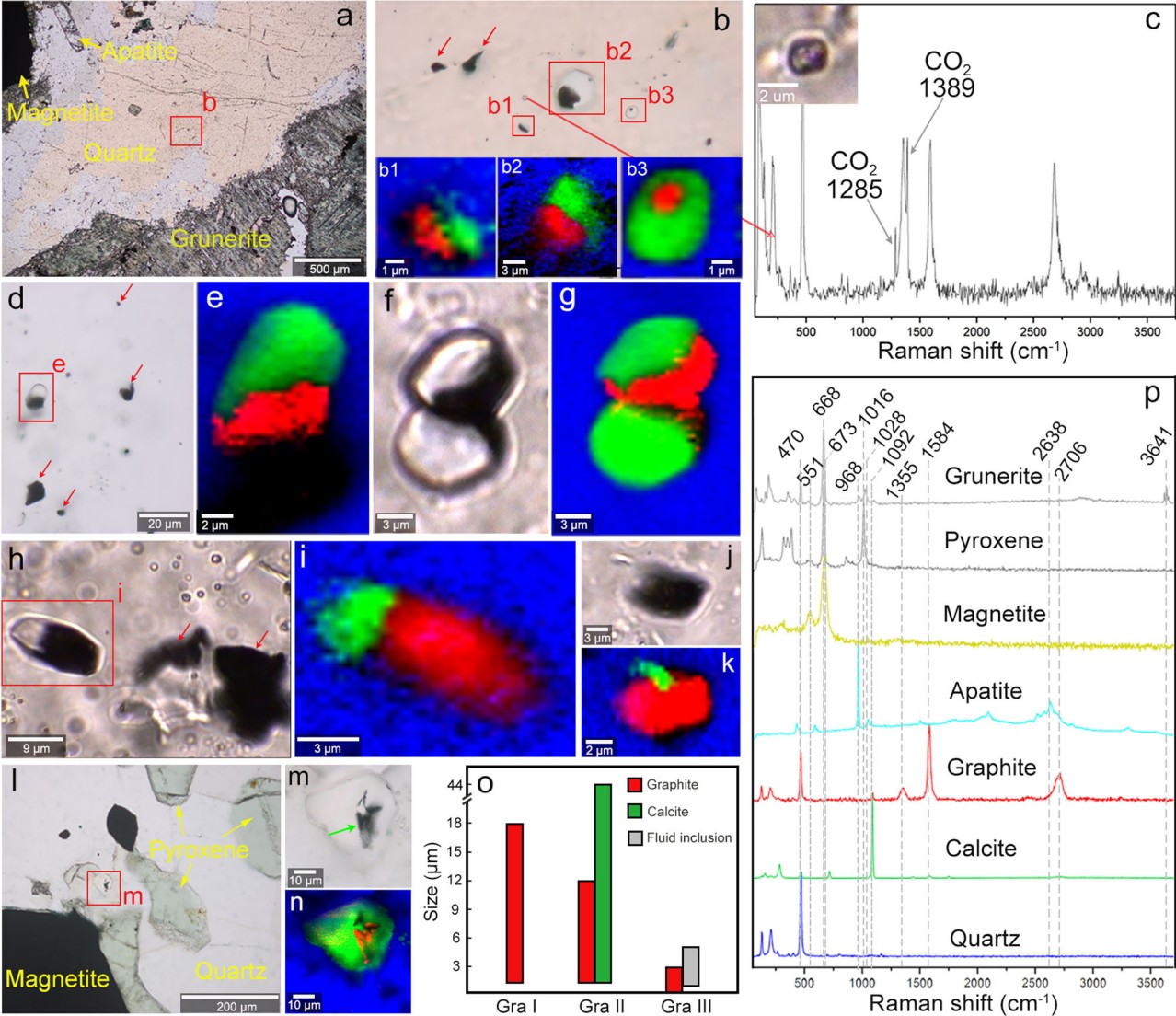

**Fig. 2 | Three petrographic types of graphite in the Saglek-Hebron banded iron formation (BIF), *SG-274* and *SG-236*. a** Plane Polarised Light (PPL) image showing occurrences of grunerite, quartz, magnetite and apatite in *SG-274*. **b** PPL image of three petrographic types of graphite including pure graphite inclusions (Gra$_I$; red arrows), calcite + graphite associations (Gra$_{II}$; b1-b3) and graphite associated with C-H-O fluid inclusions (Gra$_{III}$). **c** Raman spectrum of Gra$_{III}$ in **b** with an inset of its image. **d** PPL image of Gra$_{II}$ surrounded by Gra$_I$ (red arrows). **e** Raman map of Gra$_{II}$ in panel **d**. **f** Transmitted Light (TL) image of twins of Gra$_{II}$. **g** Raman map of panel **f**.

**h** TL image showing occurrences of Gra$_{II}$ and Gra$_I$. **i** Raman map of Gra$_{II}$ in panel h. **j** TL image of carbonate almost entirely encased in graphite. **k** Raman map of panel **j**. Note that the area ratio of graphite to calcite in two dimensions is increasing from ~5% to 97% from panel **b** to panel **k**. **l** PPL image showing occurrences of pyroxene, quartz and magnetite in *SG-236*. **m** PPL image of Gra$_{II}$ with nanometre thick books of graphene (green arrow) in panel **l**. **n** Raman map of panel m. **o** Histogram of the size ranges of the Gra$_I$, Gra$_{II}$ and Gra$_{III}$. **p** Raman spectra for this figure. Raman map colours: red – graphite, blue – quartz, green – calcite, and black – poor signal.

olivine + antigorite + magnetite and is best described as a marble (Fig. 1f and Supplementary Data 1). Mineralogical maps obtained by energy dispersive X-ray spectroscopy (EDS) show that the constituent mineral phases intermix on micron-scale, particularly quartz and magnetite, and magnetite grains are primarily hosted in the grunerite-, olivine- and pyroxene-rich bands in samples *SG-274*, *SG-275* and *SG-236*, respectively (Fig. 1d–f). This mineral paragenesis represents a typical prograde product under amphibolite-facies metamorphism[26,27]. Secondary, retrograde chlorite is usually pale green and intrudes the quartz-rich band as linear array from the rims of grunerite-rich band in sample *SG-274*, representing a prevalent retrograde mineral in these amphibolite-grade BIF (Supplementary Fig. 2). In the selected area of *SG-274*, 640 calcite grains were identified in quartz-rich band using polarising microscopy and micro-Raman spectroscopy, among which 64 (10%) were associated with graphite (Fig. 1g). This proportion is notably lower than those for apatite + graphite associations in the

Eoarchean Akilia quartz-pyroxene rock (about 20%)[11] and Nuvvua-gittuq Fe-silicate BIF (~25%)[28]. In the above rocks, the graphite grain sizes vary between 1 and 45 μm and they commonly occur as flakes coating apatite crystals that vary between 3 and 100 μm in size. They also occasionally occur in other mineral assemblages, such as hornblende ± calcite ± magnetite ± pyrrhotite ± chalcopyrite, which are often distributed near the edge of large hornblende grains[11]. In comparison, graphite in the Saglek-Hebron BIF is generally smaller in size and varies between ~1 and 12 μm, and mainly occurs as thin coatings on microscopic calcite grains inside millimetre-size quartz crystals (Fig. 2).

Graphitic carbons in the Saglek-Hebron metasedimentary rocks can be grouped into four types based on their textural features, petrographic mode of occurrence and associated minerals. The first petrographic type of graphite (Gra$_I$) is pure solid inclusion disseminated within coarse quartz grains, while the second (Gra$_{II}$), third (Gra$_{III}$) and

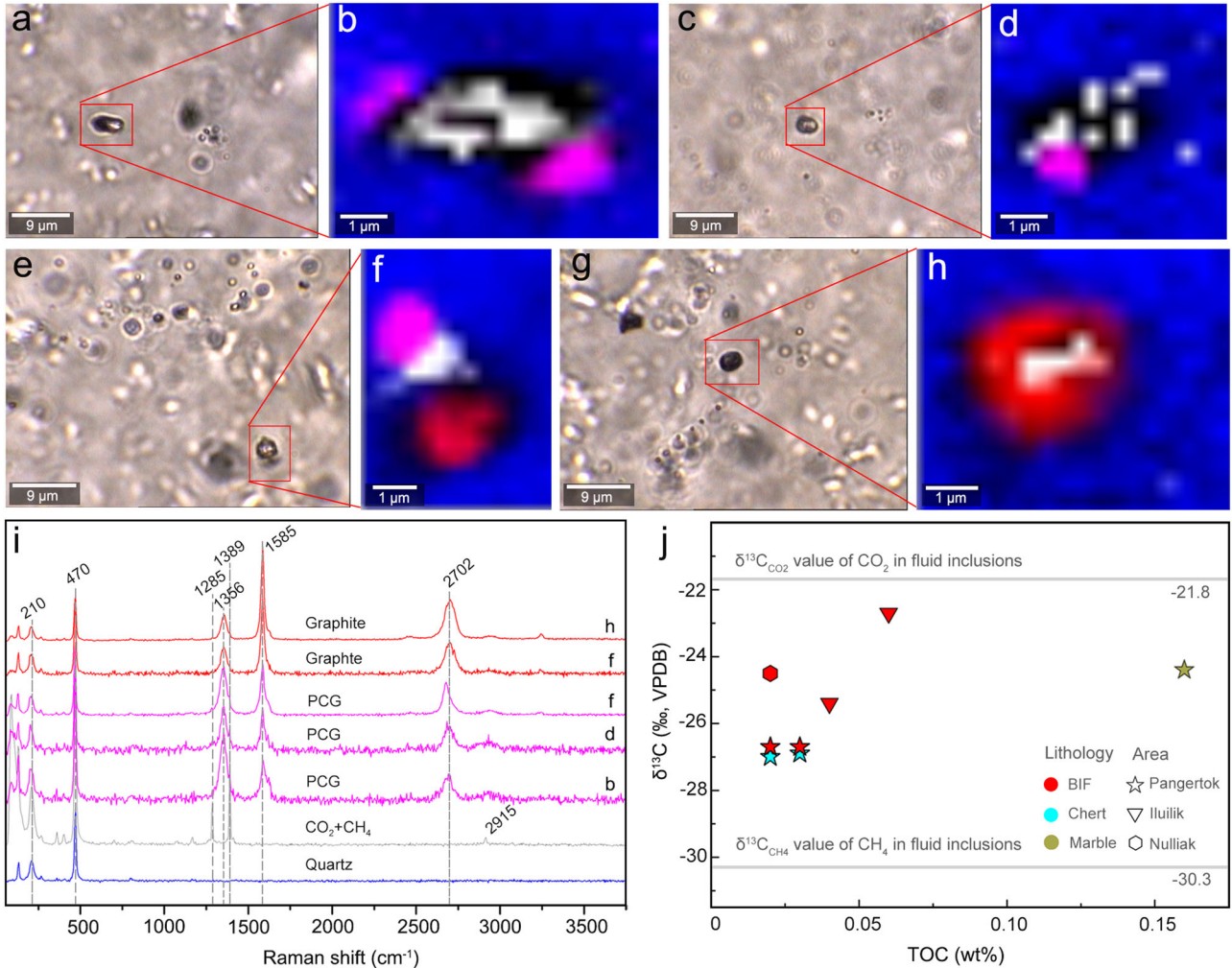

**Fig. 3 | Fluid inclusions in the Saglek-Hebron banded iron formation (BIF), SG-274.** Transmitted Light (TL) images (**a**, **c**) and corresponding Raman maps (**b**, **d**) of fluid inclusions containing $CO_2 + CH_4$ and poorly crystalline graphite (PCG). **e** TL image of fluid inclusion containing $CO_2 + CH_4$, PCG and graphite. **f** Raman map of fluid inclusion in panel **e**. **g** TL image of fluid inclusion containing $CO_2 + CH_4$ and graphite. **h** Raman map of fluid inclusion in panel **g**. **i** Raman spectra for this figure, showing increasing crystallinities of graphite from panel **b** to panel **h**. Raman map colours: red – graphite, pink – PCG, blue – quartz, grey – $CO_2 + CH_4$, and black – poor signal. **j** Plot of $\delta^{13}C$ versus TOC contents, illustrating that $\delta^{13}C$ values of graphite fall between $CO_2$ and $CH_4$ in fluid inclusions.

fourth ($Gra_{IV}$) petrographic types of graphite are closely associated with calcite grains, C-H-O fluid inclusions, and magnetite, respectively (Figs. 2–4). In the Saglek-Hebron BIF, the $Gra_I$, $Gra_{II}$ and $Gra_{III}$ co-occur almost entirely within the central zone of the quartz-rich band, while pure calcite grains without graphite are mainly enriched near the edge of coarse quartz and contact with millimetre size magnetite and grunerite (Fig. 1g). This indicates that graphite could have been oxidised by trivalent iron into $CO_2$ from which calcite formed during retrograde metamorphism, which is consistent with the lower proportion of calcite + graphite associations than those of apatite + graphite associations[11,28]. The $Gra_I$ disseminated within the centre of the quartz-rich band commonly has irregular or deformed morphologies with sizes ranging from ~1 to 18 μm (Fig. 2b, d, h, o), which may be derived from the infringement on the surrounding minerals during pre- to syn-metamorphic graphite crystallisation[29,30]. The $Gra_{II}$ is flaky in shape with slightly smaller sizes (~1 to 12 μm) relative to the $Gra_I$, and typically appears as thin coatings on ellipsoidal to subhedral calcite grains with area ratios ranging from ~5% to 97% in two dimensions (Fig. 2b, d–o). The $Gra_{II}$ is always mixed with $Gra_I$, which is significantly more abundant than $Gra_{II}$, and it is usually surrounded by fluid inclusions within the centre of the quartz-rich band (Fig. 2b, d–n and Supplementary Fig. 3). The $Gra_{III}$ (<3 μm) associated with small micron-sized fluid

inclusions (0.7–5.0 μm) are commonly dark in appearance within coarse quartz grains (Figs. 2c, o and 3 and Supplementary Fig. 3). In comparison, the $Gra_{IV}$ in the Saglek-Hebron marble occurs as micrometre size opaque patches closely intertwined with magnetite within and on the edges of carbonate grains (Fig. 4b–i), while it can also display a filamentous texture with ~24 μm in length (Fig. 4j, k). In addition, a large number of graphite-free, primary fluid inclusions that occasionally contain opaque material, occur as trails or linear arrays within coarse quartz grains in the Saglek-Hebron BIF (Supplementary Fig. 3).

Micro-Raman spectroscopy reveals that the four petrographic types of graphite found in the Saglek-Hebron metasedimentary rocks display spectral features varying from poorly to well crystalline graphite, including a narrow and sharp G band between 1583 and 1589 $cm^{-1}$ (FWHM = 18–48 $cm^{-1}$), a narrow, sharp, but variable-intensity D band between 1352 and 1368 $cm^{-1}$ (FWHM = 21 to 80 $cm^{-1}$), and a 2D band between 2683 and 2714 $cm^{-1}$ (FWHM = 45–86 $cm^{-1}$; Supplementary Data 2 and Supplementary Fig. 4). Notably, most $Gra_{III}$ have stronger D- and 2D-bands intensities and lower frequency 2D band (from 2683 to 2703 $cm^{-1}$) relative to other petrographic types of graphite (Supplementary Fig. 4c, d), while the $Gra_{IV}$ shows more variable and larger D- and G-bands FWHM (Supplementary Fig. 4a, b). The

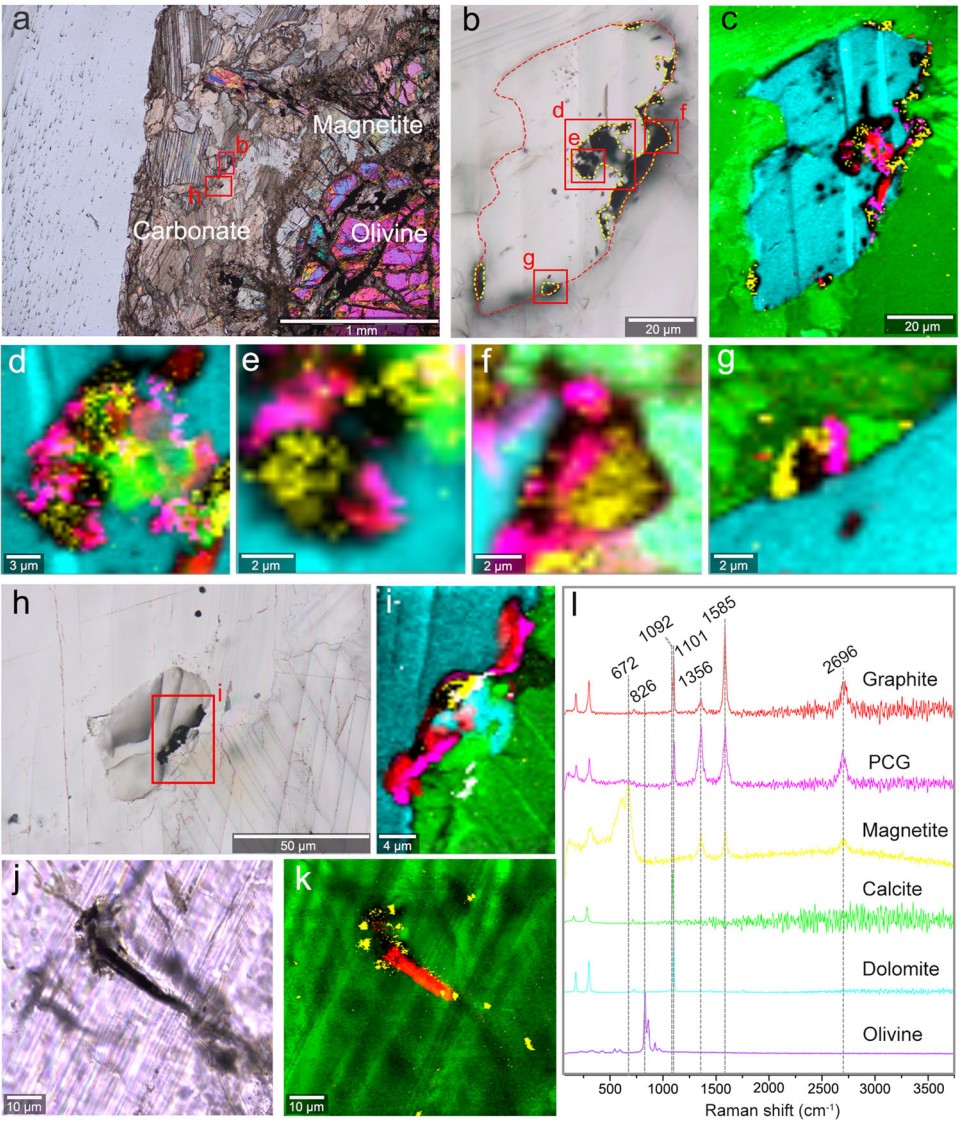

**Fig. 4 | Graphitic carbon associations with magnetite (Gra$_{IV}$) in the Saglek-Hebron marble. a** Cross polar image showing occurrences of carbonate, magnetite and olivine with serpentine veins. **b** Plane Polarised Light (PPL) image of an intra-grain cavity and arborescent coatings of opaque phases (yellow dotted contour line) within the dolomite grain (red dotted contour line). **c** Raman image of panel **b** showing the reaction products of decarbonation reactions within the cavity including micro- to nanometre size disseminations of graphite, poorly crystalline graphite (PCG), and magnetite. **d**–**g** Enlarged views showing the close association of variably crystalline graphite with magnetite. **h, i** PPL and Raman images of an intra-grain cavity and coatings of opaque phases including micro- to nanometre size disseminations of graphite, PCG, and magnetite. **j, k** Transmitted Light and Raman images of a filamentous graphite structure associated with disseminations of micrometre size magnetite in fracture of a calcite grain. **l** Raman spectra for the Saglek-Hebron marble. Raman map colours: red – graphite, pink – PCG, green – calcite, turquoise – dolomite, yellow – magnetite, and black – poor signal.

occurrence of 2D band collected from graphite in the Saglek-Hebron metasedimentary rocks is markedly different from kerogen's three peaks in the 2600–3100 cm$^{-1}$ region, potentially excluding younger infiltrations from elsewhere[31]. The 2D band may represent turbostratic or curled graphite structures, while D and G bands of graphite represent disordered and ordered graphitic carbon, respectively[28,32].

In the Saglek-Hebron metasedimentary rocks, the 2D/G peak intensity ratios for the Gra$_I$, Gra$_{II}$, Gra$_{III}$ and Gra$_{IV}$ range from 0.22 to 0.67, from 0.29 to 0.83, from 0.44 to 0.97, and from 0.25 to 0.70, respectively, whereas the D/G peak intensity ratios for the Gra$_I$, Gra$_{II}$, Gra$_{III}$ and Gra$_{IV}$ range from 0.04 to 0.78, from 0.05 to 0.80, from 0.30 to 1.70, and from 0.09 to 1.09, respectively (Supplementary Data 2 and Supplementary Fig. 4d). The large difference in the D/G peak intensity ratios is derived from their heterogeneity in crystallinity rather than from artefacts caused by polishing and graphene sheet orientation[33], considering that all graphite was analysed below the thin section surface[27,34]. The enhancement of the 2D/G peak intensity ratios is similar to those of graphite "whiskers" or cones (Fig. 2m and Supplementary Fig. 4d)[35,36], which are attributed to spiral and conical growth patterns[11,37,38]. High-resolution transmission electron microscopy (HRTEM) images show graphitic lattice fringes with (002) interplanar spacings ranging from 3.23 Å to 3.84 Å (Supplementary Fig. 5b-g), among which expanded lattices larger than the 3.35 Å spacing typical for crystalline graphite are consistent with poorly crystalline graphite (PCG)[39]. The co-occurrence of multiple crystallinities of graphite has been documented in early Phanerozoic S-type granite from the Lusatian block of eastern central Europe[40], Proterozoic gneiss and quartzite from the Iberian metamorphic belt of Spain[41], Archean and Paleoproterozoic metamorphosed BIF from Nuvvuagittuq (Canada), Anshan (China), Wutai (China), and Michigamme (USA)[27,42], as well as in chert from the Apex Formation in Western Australia[43]. The degree of graphitic order has been widely used to calculate the crystallisation

**Table 1 | Stable isotopic compositions of graphite (gra) and carbonate (carb) in bulk-rock powders of metasedimentary rocks in this study**

| Sample ID | Lithology | $C_{gra}$ (wt%)[a] | $\delta^{13}C_{gra}$ (‰) | $\delta^{13}C_{carb}$ (‰) | $\delta^{13}C_{carb}$ - $\delta^{13}C_{gra}$ |
|---|---|---|---|---|---|
| SG-202 | BIF | 0.06 | −22.7 | −5.5 | 17.2 |
| SG-207 | BIF | 0.04 | −25.4 | −5.1 | 20.3 |
| SG-236 | BIF | 0.02 | −24.5 | −4.1 | 20.4 |
| SG-274 | BIF | 0.02 | −26.7 | −9.2 | 17.5 |
| SG-275 | Marble | 0.16 | −24.4 | −1.4 | 23.0 |
| SG-276 | Chert | 0.03 | −26.9 | −11.2 | 15.7 |
| SG-277 | Chert | 0.02 | −27.0 | −5.9 | 21.2 |
| SG-278 | BIF | 0.03 | −26.7 | −7.6 | 19.2 |
| Average | | | −25.5 | −6.3 | 19.3 |

[a]The carbon contents were measured in bulk-rock powders by EA-IRMS after acidification, which only removed carbonate and did not separate the kinds of graphitic carbons.

temperature of graphite under peak metamorphic conditions[12,15,28]. Based on the geothermometer using Raman spectroscopy of organic matter calibrated for temperatures between 330 °C and 641 °C[44], we calculate the crystallisation temperatures with a precision of about ±50 °C for the $Gra_I$, $Gra_{II}$, $Gra_{III}$ and $Gra_{IV}$ to be between 425 °C and 627 °C, between 415 °C and 610 °C, between 336 °C and 498 °C, and between 368 °C and 583 °C, respectively (Supplementary Data 2). The higher estimated crystallisation temperatures of graphite are consistent with upper amphibolite-facies metamorphism (>500 °C) experienced by the host rocks, potentially indicating an indigenous and syngenetic source of graphitic carbons, while the lower ones point to a secondary origin (Supplementary Fig. 4f).

The carbon isotope values of graphite ($\delta^{13}C_{gra}$) in the Saglek-Hebron metasedimentary rocks range from −27.0‰ to −22.7‰ (average = −25.5‰, $n$ = 8), while the carbon isotope values of carbonate ($\delta^{13}C_{carb}$) show an overall range of 9.8‰ (−11.2‰ to −1.4‰, average = −6.3‰, $n$ = 8; Table 1). The low and large variable $\delta^{13}C_{carb}$ values indicate that these carbonates formed from the $^{12}C$-enriched $CO_2$ produced by the oxidation of organic matter in rocks[15,42,45], as suggested by our observation on the petrographic mode of occurrence with more $Gra_I$ and $Gra_{II}$ near the central zone of quartz-rich band and more graphite-free calcite grains near the Fe-silicate + magnetite portions of the Saglek-Hebron BIF (Fig. 1g). This explanation is further supported by recent works that reveal the oxidation of rock organic carbon has released significant amounts of $CO_2$ that offsets the $CO_2$ drawdown by silicate weathering at the global scale[46,47]. The negative $\delta^{13}C_{gra}$ values in the Saglek-Hebron metasedimentary rocks fall well within the range of $\delta^{13}C$ values reported in the Akilia BIF (− 49.0‰ to −12.6‰)[9,10,27,38], Nuvvuagittuq BIF (−28.1‰ to −18.2‰)[27,28], Saglek-Hebron pelitic rocks (−28.2‰ to −11.0‰)[12], as well as the average composition of sedimentary organic matter over the last 3.5 Ga[48], and are consistent with biological fractionation when assuming that ancient biological processes produced isotopically light carbon deposits similar to modern biological processes. However, high-resolution elemental mappings acquired by NanoSIMS (Supplementary Fig. 6) and elemental compositions acquired by TEM-EDS (Supplementary Data 3) show that graphite in the Saglek-Hebron metasedimentary rocks is comprised of only carbon (C) and hydrogen (H), but lacks in other biologically important elements, such as nitrogen (N), phosphorus (P) and sulfur (S)[34,38,42]. Such chemical compositions, alongside the absence of apatite + graphite associations representing a common occurrence of the preserved early traces of life in BIF[27,38,42], point to abiotic processes that may produce biological signatures, such as decarbonation reaction at high temperature (≥400 °C)[49,50], Fischer-

Tropsch-type (FTT) reaction at low temperatures (i.e., surface-catalysed reduction of inorganic carbon; 200–350 °C)[51–53] and deposition from C-H-O fluids[27,28,38,40,54].

## Origin of graphite in the Saglek-Hebron BIF

The decarbonation has been considered as a thermal decomposition of Fe-carbonate ($FeCO_3 \rightarrow Fe_3O_4 + CO_2 + C$), which could produce graphite with similar negative $\delta^{13}C_{gra}$ signatures in carbonate-bearing rocks in Isua, West Greenland (e.g., −30‰)[49,55]; however, the lack of the petrographic association of the type of graphite with magnetite clearly excludes this reaction in the Saglek-Hebron BIF. Calcite reduction (or decarbonation: $CaCO_3 + SiO_2 \rightarrow CaSiO_3 + C + 2O_{fluid}$) has also been documented to produce graphite with a maximum fractionation of ~15‰ in carbonate-bearing rocks in Alpine Corsica, France[56]. However, such a fractionation is notably lower than the carbon isotopic difference between graphite and carbonate in the Saglek-Hebron BIF ($\delta^{13}C_{carb}$ - $\delta^{13}C_{gra}$ = 17.5‰ to 20.4‰; Table 1). The absence of wollastonite co-produced by calcite reduction further rules out the possibility of calcite reduction in the Saglek-Hebron BIF. The FTT reaction commonly occurs on the surface of metal catalysts in hydrothermal systems to reduce CO or $CO_2$ to relatively simple organic compounds[52,53,57], but no association of graphite with catalyst for the FTT reaction (e.g., magnetite) was found in the Saglek-Hebron BIF. The restricted range of $\delta^{13}C_{gra}$ in the Saglek-Hebron BIF is also inconsistent with FTT-derived organic compounds[58]. In fact, it has been proposed that C-H-O fluids with variable $CO_2/CH_4$ ratios can result in the deposition of graphite with $\delta^{13}C$ values ranging from −30‰ to −9‰ in metasedimentary rocks from New Hampshire[54]. Therefore, we suggest that graphite in the Saglek-Hebron BIF is most likely derived from C-H-O fluids considering two lines of evidence: 1) the associations of graphite with C-H-O fluid inclusions are widely found in quartz-rich band (Figs. 2c and 3a–i), and 2) the $\delta^{13}C$ values of graphite fall well within the range of $\delta^{13}C$ values for fluid inclusions containing mixtures of $CO_2$ and $CH_4$ ($\delta^{13}C_{CO2}$ = − 21.8‰ and $\delta^{13}C_{CH4}$ = − 30.3‰; Fig. 3j) in the Saglek-Hebron BIF. As such, the variability of $\delta^{13}C_{gra}$ values in the Saglek-Hebron BIF may arise from the change in the $CO_2/CH_4$ ratio in the C-H-O fluids during metamorphism[40,54,59].

In the Saglek-Hebron BIF, the co-occurrence of the $Gra_I$, $Gra_{II}$ and $Gra_{III}$ within micrometre distances (Fig. 1g), together with the absence of graphite veins cross-cutting the Saglek-Hebron BIF, indicates that they have a localised source and were transported together over short distance by metamorphic fluids. The retrograde minerals (such as chlorite; Supplementary Fig. 2) in the Saglek-Hebron BIF suggests the cooling of C-H-O fluids during retrograde metamorphism, which can induce the hydration reactions of the host minerals and decreased carbon solubility, and subsequently result in the precipitation of PCG via the reduction of $CO_2$ and the oxidation of $CH_4$ in aqueous fluids ($CO_2 + CH_4 \rightarrow 2C + 2H_2O$)[27,28]. This model is supported by TEM-EDS images that show the close association of nano-chlorite with PCG (Supplementary Fig. 5h-o). This process has been documented in mafic and ultramafic rock units of the deep-drilling cores in Germany[60], granulites from southern India[61], as well as basalt and pyroxenite on Mars[62], and recently modelled by Rayleigh fractionation equation to confirm that graphite precipitated from a $CO_2$-$CH_4$ fluid could have $\delta^{13}C$ values ranging from −44.6‰ to −7.5‰[40,54]. During the cooling, increasing $CO_2$ disassociation in metamorphic fluids could result in the synchronous precipitation of calcite and graphitic carbon, potentially producing the observed close association of calcite with graphite (Fig. 2)[11]. The geological $CH_4$ is generally thought to come from three sources[63]: microbial origin ($\delta^{13}C_{CH4}$ < − 50‰; emitted by methanogenic archaea), thermogenic origin ($\delta^{13}C_{CH4}$ = − 50‰ to −20‰; produced by the thermal decomposition of organic matter), and abiogenic origin ($\delta^{13}C_{CH4}$ > − 20‰; generated from the mantle or non-biological reactions). The $\delta^{13}C_{CH4}$ value of $CH_4$ ($\delta^{13}C_{CH4}$ = − 30.3‰) in fluid inclusions falls well within the range of carbon isotopic compositions for

thermogenic $CH_4$, revealing that the $CH_4$ could be a product of thermal decomposition of organic matter in the Saglek-Hebron BIF. The $\delta^{13}C_{CO2}$ value of $CO_2$ ($\delta^{13}C_{CO2} = -21.8‰$) in fluid inclusions is typical of that produced by the oxidation of such $CH_4$ rather than carbonates and igneous carbon of mantle origin due to these two carbon sources having $\delta^{13}C$ values heavier than $-14‰$[64–66]. Therefore, the reactants $CO_2$ and $CH_4$ for the fluid-deposition of PCG could be in-situ liberated from a common syngenetic source of microbial organic matter[54], which occurs as crystalline graphite in the Saglek-Hebron BIF[27]. Such an organically derived C-H-O fluid has been known to result in the deposition of graphite with $\delta^{13}C$ values ranging from $-30.6$ to $-20‰$ in volcanic rocks of the external zone of the Betic Cordillera, Southern Spain[67], basalts and black shales in British Columbia, Canada[68], andesites in Borrowdale, England[59], metagabbros and amphibolites of the deep-drilling cores in Germany[60], as well as Water Hen intrusion in the Duluth Complex, USA[64]. This indicates that PCG was deposited from C-H-O fluids derived from the retrograde cracking and oxidation of residual organic matter while crystalline graphite stemmed from high-grade metamorphism of syngenetic organic matter. The petrographic observations, nanoscale features and variable peak crystallisation temperatures of graphite in the Saglek-Hebron BIF provide further support for this conclusion (Fig. 1g and Supplementary Data 2). The varying crystallinity of graphitic carbons in the Saglek-Hebron BIF, thus, are sourced from the preservation of fluid-deposited graphite along with syngenetic graphite[27,41], which point to primitive organic matter with low $\delta^{13}C$ values during prograde and retrograde metamorphism.

### Origin of graphite in the Saglek-Hebron marble

In the Saglek-Hebron marble, we observed the Fe-carbonate + graphite + magnetite associations rather than fluid inclusions (Fig. 4 and Supplementary Fig. 7), providing a plausible indication for the occurrences of decarbonation and FTT reactions. Petrographic observations suggest that serpentinization reactions (olivine + water => serpentine + magnetite + hydrogen) in the Saglek-Hebron marble possibly have provided a source of $H_2$, CO and catalyst for the FTT or Friedel–Crafts-type reactions linked to serpentinization[51,69,70]. However, magnetite within serpentine veins is not found to be associated with graphite in the Saglek-Hebron marble (Supplementary Fig. 8), and the FTT reaction is known to be most effective between 200 and 350 °C to generate methane and short chain hydrocarbons rather than structurally complex graphite[12,32,52,53,57,71]. Also, such reaction has never been conclusively confirmed to play a significant role in formation of graphite in nature[72]. Alternatively, the intertwinement of graphite to magnetite within Fe-carbonate grains is most likely a product of decomposition of Fe-carbonate (Fig. 4)[49,73], which can be expressed as the following equation[74]: Fe-dolomite + water => Mg-calcite + magnetite + graphite + methane. High-resolution Raman hyperspectral images reveal that the $Gra_{IV}$ in the Saglek-Hebron marble is closely associated with magnetite and calcite and specifically within Fe-dolomite grains (Fig. 4). Such mineral assemblage has been verified to indicate formation of graphite from decarbonation in subduction zone settings by petrological observation and experimental simulation[73], as implied by the filamentous graphite in the Saglek-Hebron marble that is comparable to graphite cones and nanotubes from the Kola Peninsula[37] and filamentous carbon synthesised by high-temperature, high-pressure treatment under hydrothermal conditions[75]. Furthermore, the fact that graphite occurs exclusively in association with magnetite within Fe-bearing carbonate grains rather than magnetite within serpentine veins in the Saglek-Hebron marble provides further support for this conclusion (Fig. 4 and Supplementary Figs. 7 and 8). Therefore, our petrographic, spectroscopic and geochemical evidence clearly support abiotic decarbonation reactions as responsible for the formation of graphite in the Saglek-Hebron marble.

In conclusion, we investigate the specific mineral associations with graphitic carbons in BIF and marble from the Saglek-Hebron Gneiss Complex, Canada, and find that graphitic carbons in the Saglek-Hebron BIF originate from metamorphism of syngenetic organic matter and C-H-O fluid deposition during retrograde metamorphism, while graphitic carbons in the Saglek-Hebron marble are ascribed to the decomposition of Fe-carbonate. In this study, a suite of converging and mutually supportive evidence indicative of abiotic synthesis of organic matter was established for graphitic carbons in the Saglek-Hebron metasedimentary rocks, which extends knowledge of the origins of organic matter in early Archean rocks. Collectively, our findings provide solid support for previous interpretations that graphite could be precipitated from C-H-O fluids derived from the retrograde cracking and oxidation of syngenetic biomass[27,28], and also provide the oldest evidence for abiotic synthesis of graphitic carbons via decarbonation in the Saglek-Hebron metasedimentary rocks. Therefore, this study could pave the way for deciphering the poorly understood effects of metamorphism on prebiotic organic matter and biogeochemical cycling of carbon, as well as the geological transformation of precursor biomass on early Earth.

## Methods

### Optical microscopy

Graphite was located within the polished thin sections using a BX-51 Olympus microscope in the Department of Earth Sciences at University College London (UCL) equipped with transmitted and reflected light. No immersion oil was used to map petrographic and mineralogical features in thin sections. The thin sections were prepared according to conventional techniques with a final 0.25 mm alumina polish and cleaned with an acetone wetted kimwipe.

### TESCAN integrated mineral analyzer

Mineralogical, modal and textural analysis was performed on gold-coated thin sections using a TESCAN Integrated Mineral Analyzer (TIMA FEG3 LMH 119-0205) under liberation analysis mode[76]. The instrument was operated at an acceleration voltage of 25 kV and a beam intensity of 18.7 mA with a working distance of 15 mm to obtain back-scattered electron (BSE) image and energy dispersive X-ray spectroscopy (EDS) data with point spacings of 2.5 μm and 7.5 μm, respectively. The BSE signal intensity was calibrated on a platinum Faraday cup using the automated procedure, while manganese standard was used for checking EDS performance. The segments (i.e., mineral grains) were produced by merging areas of coherent BSE and EDS data. Individual spectra from points within each segment are summed, and then compared against a classification scheme to identify minerals in thin sections.

### Micro-Raman spectroscopy

Micro-Raman spectroscopy was performed with a WITec alpha 300 confocal Raman imaging microscope at the UCL, using a 532 nm wavelength laser and 8 mW power[5]. Raman spectra and hyperspectral scans were performed with an 100X objective (N.A. = 0.9) using an optical fibre of 50 μm in diameter, to analyze targets below the polished thin section surface, which was meant to minimise spectral artifacts caused by polishing. Hyperspectral images were produced by the main peak intensity mapping for specific minerals, which include quartz (~465 $cm^{-1}$), carbonate (~1092 $cm^{-1}$), apatite (~968 $cm^{-1}$), magnetite (~673 $cm^{-1}$), crystalline graphite (~1585 $cm^{-1}$), PCG (~1355 $cm^{-1}$), and $CO_2$ (~1389 $cm^{-1}$). Average spectra were corrected with a background subtraction of a polynomial function. Raman spectrum parameters, such as peak positions, Full Width at Half Maximum (FWHM), and areas under the curve were extracted from the background-subtracted spectra, and then modelled with Lorentz-fitted curves. Based on these parameters, the peak crystallisation temperatures of graphitic carbon in the Saglek-Hebron metasedimentary rocks can be

calculated using following geothermometer[44]:

$$T(°C) = -455 \times D1/(D1 + G + D2) + 641$$

This equation that takes into account peak-broadening effects has been calibrated against metamorphic mineral assemblages in metapelites exposed to the temperature range of 330–641 °C with the error of ±50 °C.

## Isotope ratio mass spectrometry

Analyses of carbon isotope compositions of total graphite were carried out on bulk rock powders with a Thermo-Finnigan Flash 1112 EA connected to a Thermo Delta V Isotope Ratio Mass Spectrometer via a Conflo IV gas distribution system in the Bloomsbury Environmental Isotope Facility at UCL[27]. Prior to analyses, the inorganic carbon component was removed with ultra pure 6 N HCl (Sequanal Grade, Pierce). A suite of standard materials that have $\delta^{13}C$ values ranging from −26‰ to −11‰, was analysed multiple times through each run to ensure reproducibility better than 0.2‰. Results are quoted as standard δ-notation (in units of per mil, ‰), relative to the Vienna Pee Dee Belemnite (VPDB) standard. Analyses of bulk rock powders for carbonate were performed on a Finnigan MAT 253 mass spectrometer in the Laboratory for Stable Isotope Geochemistry at the Institute of Geology and Geophysics, Chinese Academy of Sciences (IGGCAS). The $CO_2$ was extracted with pure $H_3PO_4$ at 75 °C for $\delta^{13}C$ measurements. The results are also reported against VPDB with a standard deviation better than 0.15‰.

The carbon isotopic composition of the extracted $CH_4$ and $CO_2$ in fluid inclusions was determined using a Finnigan MAT 253 mass spectrometer in the Laboratory for Stable Isotope Geochemistry at the IGGCAS. To extract fluid components from fluid inclusions in the quartz, the sample was heated under vacuum at 600 °C for 15 min and then passed through a dry ice + ethanol cold trap under vacuum to remove $H_2O$. The fluid components then passed through a liquid $N_2$ cold trap to separate $CH_4$ and freeze $CO_2$ which was subsequently analysed after heating. The remaining gases ($CH_4 + N_2$) were oxidised by reaction with CuO at 780 °C to the reacted gases ($CO_2 + H_2O + N_2$), which subsequently passed through dry ice + ethanol and liquid $N_2$ cold traps under vacuum to collect pure $CO_2$ representing $CH_4$ carbon composition of fluid inclusions. Carbon isotope compositions of $CH_4$ and $CO_2$ in the fluid inclusions are expressed as $\delta^{13}C_{CH4}$ and $\delta^{13}C_{CO2}$ relative to VPDB with a standard deviation better than ±0.024‰ and ±0.037‰, respectively.

## Scanning electron microscopy

The morphology and composition of selected targets were characterised by a JEOL JSM-6480L scanning electron microscopy (SEM) equipped with an energy dispersive X-ray spectroscopy (EDS) in the Department of Earth Sciences at UCL. The SEM was operated at standard operating conditions, which include a 15 kV accelerating voltage, 1 nA electron beam current and a working distance of 10 mm.

## Focused ion beam milling-transmission electron microscopy

The focused ion beam (FIB) milling was conducted with a Helios 5 CX FIB instrument in the Department of Materials, Imperial College London. In brief, the selected graphite-carbonate was located within the dual-beam FIB by using SEM imaging and a protective Pt shield was deposited on the surface. Then, the material from around the deposited Pt mask was sputtered away by a focused 30 kV Ga+ primary beam, making a 1 μm thick vertical section. After the bottom and sides of the section were removed, it was carefully transferred and welded onto a Cu TEM half-grid. Finally, the section was thinned down to ~100 nm with a 43 pA focused beam of Ga+. High-resolution TEM data were obtained using a JEOL2100 transmission electron microscopy operating at 200 kV in the Department of Chemistry at UCL.

## Nano-scale secondary ion mass spectrometry

Elemental maps were obtained on the polished 30-μm-thick thin sections using a CAMECA NanoSIMS 50 L (Paris, France) at the State Key Laboratory of Biogeology and Environmental Geology, China University of Geosciences (Wuhan)[34]. Each region of interest was pre-sputtered with a 1.5 nA or 3 nA beam current and an ion dose of $n = 5 \times 10^{16}$ ions/cm² to remove the conductive coating (Au) and the contaminants that were generated during pretreatment. In the multi-collection mode, negative secondary ions ($^{12}C^-$, $^{12}C^1H^-$, $^{12}C^{14}N^-$, $^{32}O^-$, $^{31}P^-$, and $^{32}S^-$) were sputtered from the sample surface using a Cs+ primary beam with ~1.6 pA intensity. During these analyses, electron gun was used for charge compensation. All of the images referred to in this paper are 512 × 512 pixels, recorded in ~24 min, and processed using CAMECA WINimage software (Version 4.4).

## Data availability

All data in this study are included in this published article and supplementary information files, and are available from the corresponding author upon request.

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

## Acknowledgements

We acknowledge the Nunatsiavut Government and Parks Canada for the permission to work in the Saglek-Hebron area. We thank J. Davy for preparing polished thin sections, E. Ware for help with FIB-SEM nano-fabrication, H. Li for carbon isotope analyses of fluid inclusions and carbonate, A.L. Jourdan for isotope analyses of total graphite, X. Xia for help with TEM analyses, and Z. Zheng for help with the NanoSIMS. Z. Guo was supported by the China Scholarship Council (CSC). This work was also funded by the National Natural Science Foundation of China (42206045 to Z.G., 42172337 to Z.S. and D.P., 42488201 and 41930322 to Z.Q.C., and 42302347 to X.Q.). Fieldwork and sample collection were supported by the Natural Sciences and Engineering Research Council of Canada grants to J.O. (RGPIN 435589-2013) and H.R. (RGPIN-2015-03982 and RGPIN-477144-2015).

## Author contributions

Z.G. and D.P. conceived the research. J.O. and H.R. participated in fieldwork. Z.G. and D.P. performed laboratory analyses. Z.G., D.P., Z.S., J.O., H.R., Z.Q.C., and X.Q. acquired the funding that supported this work. The paper was written by Z.G. and edited by D.P. with contributions from all co-authors.

## Competing interests

The authors declare no competing interests.
