## [Peer Review File · Nature Communications]

Abiotic synthesis of graphitic carbons in the Eoarchean Saglek-Hebron metasedimentary rocksREVIEWER COMMENTS

Reviewer #1 (Remarks to the Author):

Overall, the paper appears interesting, but it needs significant revisions before publication. The paper provides insights into the origin of graphitic carbon in some of the oldest rocks on Earth. However, the literature is not appropriately cited, and there are technical aspects that should be clarified. In addition, this study builds upon numerous prior investigations, some of which have already documented the variation in crystallinity of graphitic carbons in metasedimentary rocks from the Nuvvuagittuq Supracrustal Belt. The originality of the paper needs to be clarified.

The title needs to be modified. "Abiotic synthesis of biosignatures" makes no sense. A biosignature originates from biological processes. If it does not, it is abiotic. It cannot be both.

The authors selectively cite some previous work and inappropriately ignore opposing views. The points listed below should be addressed in the main text:

- L. 19-22; 58-60: The Eoarchean age of the rocks studied by Tashiro et al. (2017) has been questioned (Whitehouse et al., 2019).

“Unambiguous field relationships between ca. 3.9 Ga tonalitic gneiss and the graphite-bearing metasediments have not been demonstrated in the literature that the Saglek claim relies upon, and earlier U-Pb-Hf isotopic studies on zircon from metasediments at one of the localities used in the claim indicate a Mesoarchean to Neoarchean time of deposition.” (Whitehouse et al. 2019).

The authors treat this aspect in a superficial and incorrect way in the supplementary information. This should be addressed in the main text.

Similarly, the authors fail to acknowledge prior research that has raised doubts about the age and origin of graphitic carbon in these rocks (Alleon and Summons, 2019): “Recently, additional evidence for Eoarchean life has been claimed based on the presence of ^{13}C -depleted graphite particles within metasedimentary rocks from Isua [33] and from the Saglek block in northern Labrador, Canada [34]. Similarly, however, evidence for the antiquity of graphite is lacking. The geological history of these rocks is complex and poorly resolved. All known sequences from these locations have experienced at least two metamorphic events since their formation, the most recent reaching at least the upper amphibolite facies around 2.8–2.5 Ga [17–22]. Graphite could thus have precipitated from hydrothermal and metasomatic fluids at any time during post-Eoarchean metamorphic episodes and, at least partially, could be as young as 2.8–2.5 Ga”.

And last, but certainly not least, Alleon and Summons (2019) argued that “sedimentary graphite, irrespective of its isotopic composition, does not constitute a reliable biosignature because the rocks in which it is found are generally metamorphosed to the point where convincing signs of life have been erased.”

- L. 47-48: “purported microfossils from highly metamorphosed rocks have only been reported from

Nuvvuagittuq Supracrustal Belt (3.75 to 4.3 Ga (gigayears ago))^{4,5}

This is not true. Purported microfossils were reported in highly metamorphosed rocks from the Isua Supracrustal Belt by Pflug and Jaeschke-Boyer (1979), which were reinterpreted to be abiotic by Bridgwater et al. (1981) and Roedder (1981).

Furthermore, the origin of the purported microfossils from Nuvvuagittuq has been questioned (McMahon, 2019). This should be acknowledged in the main text.

- L. 60-63: “Hence, in this context, as well as in the context of exobiology, the distinction between abiotic and biological processes and sources of organic molecules in graphitic carbons is an imperative goal to achieve accurate identification of traces of early life, prebiotic chemistry and extraterrestrial life”.

This statement seems to me to be excessive. I am skeptical about the significance of differentiating "between abiotic and biological processes and sources of organic molecules in graphitic carbons" in the context of exobiology or prebiotic chemistry. The search for evidence of extraterrestrial life is unlikely to be carried out on rocks that have been heavily metamorphosed. This also applies to prebiotic chemistry. Could the authors provide further clarification on these aspects? Without this, exobiology and prebiotic chemistry should not be highlighted.

- Raman identification of CH₄ associated with graphitic carbon is not convincing. Assigning the band around 2915 cm⁻¹ uniquely to CH₄ is inappropriate for disordered graphitic carbons. The presence of disorder (defects) activates a broad band around 2900 cm⁻¹, which is usually attributed to D + D' or D + G" combination modes (e.g., Pimenta et al., 2007; Beyssac and Lazzeri, 2012).

The Raman spectra presented here are too noisy to distinguish a CH₄ contribution from the purely structural defects expected in disordered graphitic carbons.

- L. 147-148: “Based on the Beyssac geothermometer calibrated for temperatures between 330° C and 641° C, we calculate the crystallization temperatures with a precision of about ±50° C”.

The commonly used term for this geothermometer is Raman spectroscopy of carbonaceous material (RSCM) thermometry, rather than the Beyssac geothermometer.

More importantly, the RSCM geothermometer developed by Beyssac et al. (2002) is calibrated for a 514 nm laser source. The defect peaks of graphitic carbons are dispersive, i.e. their frequencies and intensities relative to that for the G peak vary with the laser energy (Beyssac and Lazzeri, 2012).

“This effect has been quantified by Sato et al. (2006) and Cancado et al. (2006) and is significant as it may increase the ID/IG ratio for given samples by ≈12% by simply changing the laser energy from 2.41 eV (514 nm) to 2.33 eV (532 nm)” (Beyssac and Lazzeri, 2012). The data reported here were collected using a 532 nm laser source. The estimated temperatures presented here are thus likely biased. Geothermometers calibrated for a 532 nm laser source exist and should be used (e.g., Kouketsu et al. 2014).

- L. 272-277: “Collectively, our unprecedented findings provide the oldest evidence for primitive sedimentary organic matter in graphitic carbons preserved in the Saglek-Hebron metasedimentary rocks, and could pave the way for deciphering the poorly understood effects of metamorphism on prebiotic organic matter and biogeochemical cycling of carbon, as well as the geological

transformation of precursor biomass on early Earth, or even other habitable planets.” This is highly exaggerated. This study builds upon numerous prior investigations, some of which have already documented the variation in crystallinity of graphitic carbons in metasedimentary rocks from the Nuvvuagittuq Supracrustal Belt. “Crystalline graphite is interpreted to represent the metamorphosed product of syngenetic organic carbon deposited in BIF, while poorly crystalline graphite was precipitated from C–O–H fluids partially sourced from the syngenetic carbon, along with fluid-deposited apatite and carbonate.” Dodd et al. (2019). See also Papineau et al. (2011). The originality of the current paper needs to be clarified.

Reviewer #2 (Remarks to the Author):

Review report on the manuscript entitled "Abiotic synthesis of graphitic carbon biosignatures in the Eoarchean Saglek-Hebron metasedimentary rocks"

The study presents new data on metamorphic graphite in samples from rocks of the Saglek-Hebron Gneiss Complex. The study presents a thorough analysis of graphite using various relevant techniques to deduce the isotopic fractionation and composition and structure of the graphite and the geochemical setting in which it is found. This is then used to deduce a likely metamorphic transformation history, that in several cases involve abiotic processes. The authors argue that while the process that have transformed the carbon material was abiotic, the final state of the carbon material is only consistent with the original source material being biogenic in origin. The purported claim is then that biological material was produced sometime during the formation of the rock, 3.9 billion years ago, and then this material was transformed by abiotic metamorphic process into the composition and state observed today.

The claim of original biogenicity is primarily based on carbon isotope fractionation, assuming that ancient biological processes generated lighter carbon deposits, similar to modern biological processes. While we agree with this assumption and we think the paper represents a thorough, scientific sound treatment of the data, the assumption is not universally accepted.

An objection often raised when it comes to conclusions based primarily on carbon isotope fractionation in old rocks, is the possibility of later contamination through fluid migration or introduced otherwise, particular for this set of rocks. Nevertheless we think that the data, the analysis and the complex intricate metamorphic picture presented in the manuscript mitigates this concern.

The text is well referenced and comments on previous studies by Tashiro, T. et al. (2017) who reported similar isotope fractionations in carbon material from the same set of rocks, though the issue of later introduction of carbon was more ambiguous in those cases. Nevertheless, the manuscript presents valuable data and analysis deserving of publication.

Specific points:

1) The poorly crystalline graphite (PCG) is identified and suggested to be formed from the precipitation from C-H-O fluids. It would be interesting to see the distribution of PCG on Figure 1. Does it only occur in the fluid-associated graphite (Gra3)?

2) It is implied that the fluid + graphitic carbon represents a system where syngenetic organic matter is mobilized in C-H-O fluids and redeposited as abiotic PCG and graphite elsewhere (line 232-239), and that some of the graphite represents the primitive organic matter. Is this directly reflected in the Gra1, Gra2, Gra3 association? And does this imply that Gra1 is the graphitized 'primitive OM'? The absence of heteroatoms from the Gra1 as suggested by Supplementary Figure 6 could be due to loss following metamorphism.

3) The oxidation of graphite is loosely suggested in the text to explain the patterns of graphite-calcite associations observed in Figure 1. However the mechanism for oxidation of graphite is not explained – is the oxidation a recent happening? Or is the association between graphite and calcite better explained by synchronous precipitation of graphite and calcite during retrograde metamorphism (line 213-215)?

4) In line 180 it is stated that 'lack of magnetite + graphite association clearly excludes' the formation of graphite from decomposition of Fe-carbonates. However, there is magnetite in all the samples studied.

Fig2: I suggest changing the color of the fluid inclusion bar in subfigure O to something else (grey perhaps, to match Fig3). At first glance at the Fig2, the reader is lead to think that the blue colors in the Raman maps indicates Fluid inclusions.

Supplemental information:

FigS1: Maybe add labels on b, c and d or arrows pointing into a

FigS3: What is what here? It is not clear to me why and you can distinguish between fluid inclusions and graphite inclusions based on this figure. And the Gra I, GraII etc.? Maybe indicate G and D peak as well (or lack off).

FigS5:

Scales are impossible to see and the use of cyan in text makes it unreadable.

FigS6: Line 107: Why do you want to make this statement here?

FigS8: What is the scale on a) and b). Again Cyan text on white background is not working well. Was the Raman spectra recorded on this particular sample? I guess they where extracted from h) and i)

Reviewer #3 (Remarks to the Author):

Response to Reviewers

All of us would like to thank the editors and reviewers for your considerations about this manuscript and constructive comments. We have revised it very carefully based on all of the comments. Detailed responses are provided in a point-by-point fashion as follows.

Response to Reviewer #1 (Remarks to the Author):

Overall, the paper appears interesting, but it needs significant revisions before publication.

The paper provides insights into the origin of graphitic carbon in some of the oldest rocks on Earth. However, the literature is not appropriately cited, and there are technical aspects that should be clarified. In addition, this study builds upon numerous prior investigations, some of which have already documented the variation in crystallinity of graphitic carbons in metasedimentary rocks from the Nuvvuagittuq Supracrustal Belt. The originality of the paper needs to be clarified.

Answer: We appreciate you for valuable comments, which greatly improved this manuscript. We have responded to your comments point by point below. In brief, our work is original because it petrographically and spectroscopically documents different origins of graphitic carbons in the Eoarchean metasedimentary rocks from the Saglek-Hebron Gneiss Complex. The novelty is now clarified in the conclusion.

The title needs to be modified. "Abiotic synthesis of biosignatures" makes no sense. A biosignature originates from biological processes. If it does not, it is abiotic. It cannot be both.

Answer: We have deleted the term "biosignatures" in the title. We would argue however that the word "biosignature" in the context of the Eoarchean should be defined as a 'possible' signature of life. There are very few unambiguous biosignatures in the geological record and "life" is also not a well-defined concept.

The authors selectively cite some previous work and inappropriately ignore opposing views. The points listed below should be addressed in the main text:

- L. 19-22; 58-60: The Eoarchean age of the rocks studied by Tashiro et al. (2017) has been questioned (Whitehouse et al., 2019).

"Unambiguous field relationships between ca. 3.9 Ga tonalitic gneiss and the graphite-bearing metasediments have not been demonstrated in the literature that the Saglek claim relies upon, and earlier U-Pb-Hf isotopic studies on zircon from metasediments at one of the localities used in the claim indicate a Mesoarchean to Neoarchean time of deposition." (Whitehouse et al. 2019).

The authors treat this aspect in a superficial and incorrect way in the supplementary information. This should be addressed in the main text.

Answer: Thank you for providing valuable references. We agree that one of the geochronology issues regarding Tashiro et al. (2017) study, is that the pre-3.95Ga age was called into question. This age comes from zircons from an orthogneiss sample, which was revised and published to be 3.92 Ga (Shimojo et al., 2016). Using the data from Shimojo et al., (2016), Whitehouse et al. (2019) argued that the age of this orthogneiss is closer to 3.865 Ga and the same outcrop was sampled and analysed by Wasilewski et al. (2021) who obtained an age of 3.869 Ga. Moreover, both Whitehouse et al. (2019) and Wasilewski et al. (2021) questioned the field relationships reported by Shimojo et al. (2016). Currently, the best age constraints for the oldest Nulliak supracrustal assemblage (which includes the samples studied here) is arguably from Sm-Nd and Lu-Hf isotopes (Morini et al., 2017; 2018) and deposition ages of associated detrital metasediments (Nutman et al., 1989) consistent

with an age of ~3.8 Ga. Hence, we are no longer relying on the age of the 3.9 Ga tonalitic gneiss to constrain the age of the metasediment samples studied here, but rather on the fact that the metasediments are part of Nulliak supracrustal exposures and therefore best constrained to be ~3.8 Ga. To make this clearer, we have cited some relevant references in the main text to include some of the debated point about the Eoarchean to Neoproterozoic ages of the Saglek-Hebron Gneiss Complex (Lines 64-73).

Similarly, the authors fail to acknowledge prior research that has raised doubts about the age and origin of graphitic carbon in these rocks (Alleon and Summons, 2019): “Recently, additional evidence for Eoarchean life has been claimed based on the presence of ^{13}C -depleted graphite particles within metasedimentary rocks from Isua [33] and from the Saglek block in northern Labrador, Canada [34]. Similarly, however, evidence for the antiquity of graphite is lacking. The geological history of these rocks is complex and poorly resolved. All known sequences from these locations have experienced at least two metamorphic events since their formation, the most recent reaching at least the upper amphibolite facies around 2.8–2.5 Ga [17–22]. Graphite could thus have precipitated from hydrothermal and metasomatic fluids at any time during post-Eoarchean metamorphic episodes and, at least partially, could be as young as 2.8–2.5 Ga”.

And last, but certainly not least, Alleon and Summons (2019) argued that “sedimentary graphite, irrespective of its isotopic composition, does not constitute a reliable biosignature because the rocks in which it is found are generally metamorphosed to the point where convincing signs of life have been erased.”.

Answer: Thank you for providing this reference and for pointing out this debate. We agree with these authors that metamorphism would result in the variations in carbon isotopic compositions of organic matter, thereby leading to blurred carbon isotopic compositions, which cannot be used alone to distinguish between abiotic and biogenic sources. However, we respectfully disagree with the quoted statement by Alleon and Summons that sedimentary graphitic carbons are always ‘metamorphosed to the point where convincing signs of life have been erased’ because 1) there are several reports in the peer-reviewed literature where researchers have used carbon isotopes and graphitic carbons to trace past signs of life including in Hadean zircons, metasedimentary rocks of various kinds, peridotitic graphite, and eclogitic diamonds, 2) this affirmation is an opinion rather than a demonstrated fact, and 3) it is defeatist and not constructive to give up on this important scientific problem because of some people’s opinion. We agree however that the graphite could possibly be younger than the host rock’s sedimentary precursor so we have cited this reference in the main text to acknowledge that these authors raised doubts about the age and origin of graphitic carbons within metasedimentary rocks from the Saglek block reported in Tashiro et al. (2017) (Lines 76-79).

- L. 47-48: “purported microfossils from highly metamorphosed rocks have only been reported from Nuvvuagittuq Supracrustal Belt (3.75 to 4.3 Ga (gigayears ago))4,5”

This is not true. Purported microfossils were reported in highly metamorphosed rocks from the Isua Supracrustal Belt by Pflug and Jaeschke-Boyer (1979), which were reinterpreted to be abiotic by Bridgwater et al. (1981) and Roedder (1981).

Furthermore, the origin of the purported microfossils from Nuvvuagittuq has been questioned (McMahon, 2019). This should be acknowledged in the main text.

Answer: Thank you for sharing valuable references. R1 is correct in pointing out that our statement

was incorrect, so we deleted the word “only” from the sentence and also added that the origin of the Nuvvuagittuq microfossils has been questioned by McMahon (2019). We feel however that it is not useful to cite all these older papers, when 1) there is consensus in the community about the abiotic or contamination origin of the purported Isua microfossils, and 2) our new work is neither about microfossils, nor about the Isua Supracrustal Belt.

- L. 60-63: “Hence, in this context, as well as in the context of exobiology, the distinction between abiotic and biological processes and sources of organic molecules in graphitic carbons is an imperative goal to achieve accurate identification of traces of early life, prebiotic chemistry and extraterrestrial life”.

This statement seems to me to be excessive. I am skeptical about the significance of differentiating "between abiotic and biological processes and sources of organic molecules in graphitic carbons" in the context of exobiology or prebiotic chemistry. The search for evidence of extraterrestrial life is unlikely to be carried out on rocks that have been heavily metamorphosed. This also applies to prebiotic chemistry. Could the authors provide further clarification on these aspects? Without this, exobiology and prebiotic chemistry should not be highlighted.

Answer: We have deleted the context of exobiology and prebiotic chemistry here to avoid confusion.

- Raman identification of CH₄ associated with graphitic carbon is not convincing. Assigning the band around 2915 cm⁻¹ uniquely to CH₄ is inappropriate for disordered graphitic carbons. The presence of disorder (defects) activates a broad band around 2900 cm⁻¹, which is usually attributed to D + D' or D + G" combination modes (e.g., Pimenta et al., 2007; Beyssac and Lazzeri, 2012).

The Raman spectra presented here are too noisy to distinguish a CH₄ contribution from the purely structural defects expected in disordered graphitic carbons.

Answer: We agree with R1 that the disordered graphitic carbon has a broad band around 2900 cm⁻¹. However, CH₄ has a very narrow band around 2915 cm⁻¹. To distinguish a CH₄ contribution from the purely structural defects in graphite, we have double-checked all our Raman spectra to identify CH₄ with a very narrow band at 2915 cm⁻¹ and accordingly modified Figs. 3 and S3 to make the CH₄ band clearer in the revised paper. We find that the narrow peak at 2915 cm⁻¹ occurs 1) only within fluid inclusions, 2) always with CO₂, and 3) not systematically associated with graphitic carbons. Hence, the presence of the narrow 2915cm⁻¹ peak is arguably consistent with the presence of methane in fluid inclusions.

- L. 147-148: “Based on the Beyssac geothermometer calibrated for temperatures between 330° C and 641° C, we calculate the crystallization temperatures with a precision of about ±50° C”.

The commonly used term for this geothermometer is Raman spectroscopy of carbonaceous material (RSCM) thermometry, rather than the Beyssac geothermometer.

More importantly, the RSCM geothermometer developed by Beyssac et al. (2002) is calibrated for a 514 nm laser source. The defect peaks of graphitic carbons are dispersive, i.e. their frequencies and intensities relative to that for the G peak vary with the laser energy (Beyssac and Lazzeri, 2012). “This effect has been quantified by Sato et al. (2006) and Cancado et al. (2006) and is significant as it may increase the ID/IG ratio for given samples by ≈12% by simply changing the laser energy from 2.41 eV (514 nm) to 2.33 eV (532 nm)” (Beyssac and Lazzeri, 2012). The data reported here were collected using a 532 nm laser source. The estimated temperatures presented here are thus likely biased. Geothermometers calibrated for a 532 nm laser source exist and should be used (e.g., Kouketsu et al. 2014).

Answer: We appreciate this constructive comment on the geothermometer. We have now selected the term ‘the geothermometer using Raman spectroscopy of organic matter’ in the revised paper. In addition, the geothermometers in Kouketsu et al. (2014) were calibrated for low-grade and medium-grade carbonaceous material, which is applicable in range of around 150–400°C. We have used these geothermometers to estimate metamorphic temperatures of Gra_{III} and Gra_{IV} (ID/IG > 0.8; please see Supplementary Data 2), which are consistent with the temperatures estimated by Beyssac et al. (2002) within uncertainties. This indicates that the small change in the laser energy does not affect the applicability of the geothermometer using Raman spectroscopy of organic matter developed by Beyssac et al. (2002), thereby causing negligible effect on the temperature ranges estimated using R2 area ratios for a large number of specimens in this study.

- L. 272-277: “Collectively, our unprecedented findings provide the oldest evidence for primitive sedimentary organic matter in graphitic carbons preserved in the Saglek-Hebron metasedimentary rocks, and could pave the way for deciphering the poorly understood effects of metamorphism on prebiotic organic matter and biogeochemical cycling of carbon, as well as the geological transformation of precursor biomass on early Earth, or even other habitable planets.”

This is highly exaggerated. This study builds upon numerous prior investigations, some of which have already documented the variation in crystallinity of graphitic carbons in metasedimentary rocks from the Nuvvuagittuq Supracrustal Belt. “Crystalline graphite is interpreted to represent the metamorphosed product of syngenetic organic carbon deposited in BIF, while poorly crystalline graphite was precipitated from C–O–H fluids partially sourced from the syngenetic carbon, along with fluid-deposited apatite and carbonate.” Dodd et al. (2019). See also Papineau et al. (2011).

The originality of the current paper needs to be clarified.

Answer: We agree with R1 and we are acutely aware that Papineau et al., 2011 and Dodd et al., 2019 reported the variation in crystallinity of graphitic carbons in metasedimentary rocks from the Nuvvuagittuq Supracrustal Belt. New observations on the Saglek-Hebron metasedimentary rocks found: 1) there are some graphite associated with fluid inclusions containing CO₂ + CH₄ (Figs. 2c and 3a-i), and 2) the δ¹³C values of graphite (δ¹³C_{Gra} = –27.0‰ to –22.7‰) fall well within the range of δ¹³C values for fluid inclusions containing mixtures of CO₂ and CH₄ (δ¹³C_{CO₂} = –21.8‰ and δ¹³C_{CH₄} = –30.3‰; Fig. 3j). These two lines of evidence lend more convincing support to the conclusion that poorly crystalline graphite could be precipitated from C–O–H fluids derived from the retrograde cracking and oxidation of sedimentary organic matter. The most important finding of this paper is that graphite could be formed from decarbonation during metamorphism, which has never been demonstrated in Precambrian rocks. To clarify the originality of this paper, we have rewritten these sentences in the revised paper (Lines 305-309).

Reviewer #2 (Remarks to the Author):

Review report on the manuscript entitled "Abiotic synthesis of graphitic carbon biosignatures in the Eoarchean Saglek-Hebron metasedimentary rocks"

The study presents new data on metamorphic graphite in samples from rocks of the Saglek-Hebron Gneiss Complex. The study presents a thorough analysis of graphite using various relevant techniques to deduce the isotopic fractionation and composition and structure of the graphite and the geochemical setting in which it is found. This is then used to deduce a likely metamorphic transformation history, that in several cases involve abiotic processes. The authors argue that while the process that have transformed the carbon material was abiotic, the final state of the carbon

material is only consistent with the original source material being biogenic in origin. The purported claim is then that biological material was produced sometime during the formation of the rock, 3.9 billion years ago, and then this material was transformed by abiotic metamorphic process into the composition and state observed today.

The claim of original biogenicity is primarily based on carbon isotope fractionation, assuming that ancient biological processes generated lighter carbon deposits, similar to modern biological processes. While we agree with this assumption and we think the paper represents a thorough, scientific sound treatment of the data, the assumption is not universally accepted. An objection often raised when it comes to conclusions based primarily on carbon isotope fractionation in old rocks, is the possibility of later contamination through fluid migration or introduced otherwise, particular for this set of rocks. Nevertheless we think that the data, the analysis and the complex intricate metamorphic picture presented in the manuscript mitigates this concern.

The text is well referenced and comments on previous studies by Tashiro, T. et al. (2017) who reported similar isotope fractionations in carbon material from the same set of rocks, though the issue of later introduction of carbon was more ambiguous in those cases. Nevertheless, the manuscript presents valuable data and analysis deserving of publication.

Answer: We thank R2 for recognizing the valuable contribution of our study. We have acknowledged and added the underlying assumption that ancient biological processes generated isotopically light carbon deposits similar to modern biological processes in the revised paper (Lines 198-200)

Specific points:

1) The poorly crystalline graphite (PCG) is identified and suggested to be formed from the precipitation from C-H-O fluids. It would be interesting to see the distribution of PCG on Figure 1. Does it only occur in the fluid-associated graphite (Gra3)?

Answer: We thank R2 for valuable suggestion, which highlighted a possibly confusing aspect. Raman images and TEM observations reveal that some graphitic carbons consist of both crystalline graphite and PCG. While most Gra_{III} is PCG, PCG also occurs in Gra_{II}. Hence, the graphite types are defined primarily by their **petrographic** modes of occurrence, rather than their crystallinity, and PCG is not readily distinguishable from crystalline graphite in all occurrences of graphitic carbons (sometimes they are mixed together). So, in this contribution, we document the petrographic types of graphite, calcite + graphite, and some PCG-bearing fluid inclusions (red circles) in Figure 1g. We have clarified this in the revised paper.

2) It is implied that the fluid + graphitic carbon represents a system where syngenetic organic matter is mobilized in C-H-O fluids and redeposited as abiotic PCG and graphite elsewhere (line 232-239), and that some of the graphite represents the primitive organic matter. Is this directly reflected in the Gra_I, Gra₂, Gra₃ association? And does this imply that Gra_I is the graphitized 'primitive OM'? The absence of heteroatoms from the Gra_I as suggested by Supplementary Figure 6 could be due to loss following metamorphism.

Answer: We agree with R2 that the Gra_I could lose heteroatoms during metamorphism. The Gra_I is mainly crystalline graphite representing the graphitized 'primitive OM', while Gra_{III} is mainly PCG representing abiotic deposition from C-H-O fluids derived from the retrograde cracking and oxidation of syngenetic organic matter.

3) The oxidation of graphite is loosely suggested in the text to explain the patterns of graphite-calcite

associations observed in Figure 1. However the mechanism for oxidation of graphite is not explained – is the oxidation a recent happening? Or is the association between graphite and calcite better explained by synchronous precipitation of graphite and calcite during retrograde metamorphism (line 213-215)?

Answer: We recognize this confusion. Observations of the petrographic mode of occurrence with more Gra_I and Gra_{II} near the central zone of quartz-rich band and more graphite-free calcite grains near the Fe-silicate + magnetite portions of the Saglek-Hebron BIF (Fig. 1g) suggest that graphite was oxidized by trivalent iron in Fe-minerals during retrograde metamorphism. So R2's second suggestion is correct in our view, and it is unlikely that this is a recent (i.e. modern) phenomenon because graphite is mostly unreactive in air. We further note that the graphite associated with calcite has variable D/G peak intensity ratios ranging from 0.05 to 0.80, which further suggest that the graphite + calcite associations formed from both the oxidation (for crystalline graphite) and synchronous precipitation of graphite and calcite from the C-H-O fluids (for PCG) during retrograde metamorphism. To make this clearer, we have added the mechanism for oxidation of graphite in the revised paper (Lines 127-128).

4) In line 180 it is stated that 'lack of magnetite + graphite association clearly excludes' the formation of graphite from decomposition of Fe-carbonates. However, there is magnetite in all the samples studied.

Answer: We agree that magnetite is ubiquitously present in all the Saglek-Hebron BIF and marble samples. However, our point is that graphite is not documented to be petrographically associated with magnetite in the Saglek-Hebron BIF (Fig. 1g). While the decomposition of Fe-carbonates would produce graphite associated with magnetite, as seen in the Saglek marble specimen (Fig. 4), this interpretation is inconsistent with the absence of magnetite + graphite association in the Saglek-Hebron BIF. We clarified this on lines 212-213.

Fig2: I suggest changing the color of the fluid inclusion bar in subfigure O to something else (grey perhaps, to match Fig3). At first glance at the Fig2, the reader is lead to think that the blue colors in the Raman maps indicates Fluid inclusions.

Answer: We thank R2 for this constructive suggestion. We have changed the color of the fluid inclusion bar to grey.

Supplemental information:

FigS1: Maybe add labels on b, c and d or arrows pointing into a

Answer: We have added labels based on your suggestion.

FigS3: What is what here? It is not clear to me why and you can distinguish between fluid inclusions and graphite inclusions based on this figure. And the Gra I, GraII etc.? Maybe indicate G and D peak as well (or lack off).

Answer: We understand the possible confusion here. Optical microscopy and Raman spectroscopy were combined to distinguish fluid inclusions from graphite in this study. To make this clearer, we have indicated Gra_I, Gra_{II}, Gra_{III}, G and D bands in the revised figure.

FigS5: Scales are impossible to see and the use of cyan in text makes it unreadable.

Answer: We have changed the colour of scales to black, and replaced cyan with red in text to make it more readable.

FigS6: Line 107: Why do you want to make this statement here?

Answer: We agree that this interpretive statement should be in the discussion. We have deleted it from this caption.

FigS8: What is the scale on a) and b). Again Cyan text on white background is not working well. Was the Raman spectra recorded on this particular sample? I guess they were extracted from h) and i)

Answer: We have corrected them to be more readable, and clarified the Raman spectra were indeed extracted from h) and i).

Response to Reviewer #3 (Remarks to the Author):

Answer: Thank you for your contribution to this peer-review process.

REVIEWERS' COMMENTS

Reviewer #1 (Remarks to the Author):

I thank the authors for improving their manuscript. The authors have addressed most of my comments, and I recommend the paper for publication, provided they address a remaining concern about the Raman data.

I have no doubt that CH₄ is associated with CO₂ in some of the fluid inclusions. However, I remain skeptical about the attribution of the "peak" at ~2915 cm⁻¹ to CH₄ when associated with graphitic carbon: the "peak" attributed to CH₄ in Fig. 2c (and Supp. Fig. 3l,n,p) is confused with noise. The signal-to-noise ratio is too low to support the claim that CH₄ is found associated with graphitic carbon. This should be removed from the figures mentioned above and from the text.

Reviewer #2 (Remarks to the Author):

We are good with general changes made to the manuscript. However, one change appear to be a bit off:

The sentence starting line 213:

however, the lack of the petrographic type of graphite associated with magnetite clearly excludes this reaction in the Saglek-Hebron BIF.

Got us confused. We suggest it gets reformulated. Maybe something like:

however, the lack of the petrographic association of the type of graphite with magnetite clearly excludes this reaction in the Saglek-Hebron BIF.

Reviewer #3 (Remarks to the Author):

Response to Reviewers

All of us would like to thank the editors and reviewers for your considerations about this manuscript and constructive comments. We have revised it very carefully based on all of the comments. Detailed responses are provided in a point-by-point fashion as follows.

Response to Reviewer #1 (Remarks to the Author):

I thank the authors for improving their manuscript. The authors have addressed most of my comments, and I recommend the paper for publication, provided they address a remaining concern about the Raman data.

I have no doubt that CH₄ is associated with CO₂ in some of the fluid inclusions. However, I remain skeptical about the attribution of the "peak" at ~2915 cm⁻¹ to CH₄ when associated with graphitic carbon: the "peak" attributed to CH₄ in Fig. 2c (and Supp. Fig. 3l,n,p) is confused with noise. The signal-to-noise ratio is too low to support the claim that CH₄ is found associated with graphitic carbon. This should be removed from the figures mentioned above and from the text.

Answer: Thank you for recommending to publish our paper. Based on your suggestion, we have revised this Raman identification of CH₄ associated with graphitic carbon. We would argue however that the narrow peak at ~2915 cm⁻¹ should be derived from CH₄ in fluid inclusions rather than noise, because 1) graphitic carbons do not have the narrow peak at ~2915 cm⁻¹, however they have this narrow peak only when associated with fluid inclusion having strong CO₂ peaks, and 2) such a CH₄ peak associated with graphitic carbons has also been identified in early Archean gneiss from Akilia (Lepland et al., 2011) and in eclogite from Western Tianshan, China (Zhang et al., 2023).

- ✧ Lepland, A. et al. Fluid-deposited graphite and its geobiological implications in early Archean gneiss from Akilia, Greenland. *Geobiology* 9, 2–9 (2011).
- ✧ Zhang, L. et al. Massive abiotic methane production in eclogite during cold subduction. *Natl. Sci. Rev.* 10, nwac207 (2023).

Response to Reviewer #2 (Remarks to the Author):

We are good with general changes made to the manuscript. However, one change appear to be a bit off:

The sentence starting line 213:

however, the lack of the petrographic type of graphite associated with magnetite clearly excludes this reaction in the Saglek-Hebron BIF.

Got us confused. We suggest it gets reformulated. Maybe something like:

however, the lack of the petrographic association of the type of graphite with magnetite clearly excludes this reaction in the Saglek-Hebron BIF.

Answer: Based on your suggestion, we have reformulated this sentence in the revised paper.

Response to Reviewer #3 (Remarks to the Author):

Answer: Thank you for your contribution to this peer-review process.